# Chasing Virus Replication and Infection: PAMP-PRR Interaction Drives Type I Interferon Production, Which in Turn Activates ISG Expression and ISGylation

**DOI:** 10.3390/v17040528

**Published:** 2025-04-04

**Authors:** Imaan Muhammad, Kaia Contes, Moses T. Bility, Qiyi Tang

**Affiliations:** Department of Microbiology, Howard University College of Medicine, Washington, DC 20059, USA; imaan.muhammad@bison.howard.edu (I.M.); kaia.contes@bison.howard.edu (K.C.); moses.bility@howard.edu (M.T.B.)

**Keywords:** antiviral, interferon (IFN), interferon-stimulated genes (ISGs), ISGylation, RNA viruses, pathogen-associated molecular patterns (PAMPs), pattern recognition receptors (PRRs)

## Abstract

The innate immune response, particularly the interferon-mediated pathway, serves as the first line of defense against viral infections. During virus infection, viral pathogen-associated molecular patterns (PAMPs) are recognized by host pattern recognition receptors (PRRs), triggering downstream signaling pathways. This leads to the activation of transcription factors like IRF3, IRF7, and NF-κB, which translocate to the nucleus and induce the production of type I interferons (IFN-α and IFN-β). Once secreted, type I interferons bind to their receptors (IFNARs) on the surfaces of infected and neighboring cells, activating the JAK-STAT pathway. This results in the formation of the ISGF3 complex (composed of STAT1, STAT2, and IRF9), which translocates to the nucleus and drives the expression of interferon-stimulated genes (ISGs). Some ISGs exert antiviral effects by directly or indirectly blocking infection and replication. Among these ISGs, ISG15 plays a crucial role in the ISGylation process, a ubiquitin-like modification that tags viral and host proteins, regulating immune responses and inhibiting viral replication. However, viruses have evolved counteractive strategies to evade ISG15-mediated immunity and ISGylation. This review first outlines the PAMP-PRR-induced pathways leading to the production of cytokines and ISGs, followed by a summary of ISGylation’s role in antiviral defense and viral evasion mechanisms targeting ISG15 and ISGYlation.

## 1. Introduction

Viral infection begins when an infectious viral particle makes contact with a permissive host cell, initiating a complex interaction between viral components and the host cell’s responsive system [1]. Once the virus enters the cell, viral products generated de novo during replication, such as viral RNA, DNA, and proteins, are recognized as foreign entities by the host’s immune mechanisms. These viral components serve as pathogen-associated molecular patterns (PAMPs) [2,3,4], signaling the presence of infection and triggering cellular antiviral responses. The detection of these viral PAMPs is a key event in activating the host’s innate immune defenses and involves multiple, highly regulated signaling pathways [4].

The diversity of cellular defense responses at least in part depends on the nature of the viral components recognized [5,6]. For example, single-stranded RNA (ssRNA), double-stranded RNA (dsRNA), viral DNA, viral proteins, viral RNA–protein complex (RNP), and viral DNA–protein complex (DNP) each stimulate distinct immune responses by binding to specialized pattern recognition receptors (PRRs) such as RIG-I, Toll-like receptors (TLRs), cGAS, and NOD-like receptors (NLRs). These PRRs are distributed across various cellular compartments, allowing immune cells to detect viral infections at different stages in the viral life cycle. Upon binding to viral PAMPs, these receptors initiate downstream signaling cascades that ultimately lead to the activation of interferon (IFN) genes, which play a pivotal role in orchestrating the antiviral response [7].

Once activated, type I interferons (IFN-α and IFN-β) are secreted and initiate the transcription of a broad array of interferon-stimulated genes (ISGs), further amplifying the antiviral state within the cell and neighboring cells [8]. The sequential interactions between PAMPs and PRRs, along with the downstream activation of IFN and ISG pathways, form the core of the innate immune response against viral infections. In this review, we will explore the detailed mechanisms by which different viral PAMPs engage PRRs, the signaling pathways they activate, and how these pathways contribute to the cellular antiviral defense, particularly focusing on the role of ISGs and ISGylation in virus replication and infection.

## 2. Virus-Triggered Generation of Type I Interferons

### 2.1. Interaction of PAMPs and PRRs

Upon viral infection, viral particle components or viral products serve as PAMPs that are recognized by PRRs in host cells, especially myeloid cells (i.e., dendritic cells, monocytes, macrophages) [5,6]. This interaction triggers a cascade of signaling pathways, leading to the activation of innate immune response. Innate immune responses defend against viral replication through a myriad of defensive molecules that are produced by and involved in different signaling pathways. Importantly, innate immune responses subsequently activate adaptive immune responses against viral antigens, which attenuates or abrogates viral replication in the host and establishes immunity. PAMPs are conserved across various species of pathogens, allowing the immune system to recognize a broad range of biological threats. Viral PAMPs contain (1) viral double-stranded RNA (dsRNA) that is produced during the replication of RNA viruses or carried by dsRNA viruses, recognized by receptors such as RIG-I and TLR3 [9]; (2) viral single-stranded RNA (ssRNA) that is common in many RNA viruses, detected by TLR7 and TLR8 [10,11,12]; and (3) viral proteins that are viral structural components like capsid proteins and envelope proteins and are recognized by TLRs, including TLR1, TLR2, TLR4, TLR6, and TLR10 [13]. PRRs are specialized receptors that detect PAMPs or DAMPs (damage-associated molecular patterns) and initiate immune responses. The four main types of PRRs involved in antiviral responses are (1) Toll-like receptors (TLRs) that are located on cell surfaces or within endosomes and recognize a variety of microbial components (e.g., nucleic acids) [14,15,16]; (2) RIG-I-like receptors (RLRs) that are cytosolic receptors and detect viral RNA, triggering an antiviral response [17,18,19]; and (3) NOD-like receptors (NLRs) that are found in the cytoplasm and sense infection-caused cellular changes such as fluctuations in ion concentrations and reactive oxygen species (ROS) [15,20,21,22,23].

### 2.2. Signaling Pathways Leading to Type I Interferon Production

Upon PAMP-PRR binding, several intracellular signaling pathways are activated, leading to the production of type I interferons (IFN-α and IFN-β). This process occurs as follows: (1) PAMP recognition: PRRs recognize viral PAMPs. TLRs (e.g., TLR3, TLR7, and TLR8) detect dsRNA and ssRNA; RLRs like RIG-I detect cytosolic viral RNA. (2) Signaling cascade activation: Once PAMPs bind to PRRs, downstream signaling pathways are initiated:Myeloid differentiation primary response gene 88 (MyD88)-Dependent Pathway [24,25,26]: Activated by all TLRs except TLR3, primarily TLR2, TLR4, TLR5, TLR7, TLR8, and TLR9. It activates kinases such as interleukin-1 receptor-associated kinases (IRAK), specifically IRAK1 and IRAK4, and tumor necrosis factor (TNF) receptor-associated factor 6 (TRAF6), Transforming growth factor-β activated kinase 1 (TAK1), and MAPKs (mitogen-activated protein kinases, e.g., JNK, p38), leading to transcription factor activation. TAK1 phosphorylates the IκB kinase (IKK) complex, which in turn phosphorylates IκB (an inhibitor of NF-κB), leading to its degradation. This allows NF-κB to translocate to the nucleus, where it promotes the transcription of pro-inflammatory cytokines such as TNF-α, IL-1β, and IL-6. The main result of the MyD88-dependent pathway is the rapid induction of pro-inflammatory cytokines (e.g., TNF-α, IL-6, IL-12), which play a vital role in the immune response to infections.Toll/IL-1R domain-containing adaptor-inducing IFN-β (TRIF)-dependent pathway [27,28,29]: Activated by TLR3 and TLR4, this pathway activates IRF3 and NF-κB, key transcription factors for interferon production. This pathway is more focused on the production of type I interferons (IFN-α/β) and is essential for antiviral responses. TLR3 detects viral dsRNA (common in RNA viruses), and TLR4 detects LPS from Gram-negative bacteria or certain viral envelope proteins. TRIF is recruited to the TIR domain of TLR3 or TLR4, initiating the TRIF-dependent signaling pathway. TRIF activates TRAF3, which plays a key role in the activation of TBK1 (TANK-binding kinase 1) and IKKε (IκB kinase epsilon). TBK1 and IKKε phosphorylate the transcription factors IRF3 and IRF7. Once phosphorylated, IRF3 and IRF7 dimerize and translocate to the nucleus. In the nucleus, IRF3 and IRF7 bind to the promoter regions of type I interferon genes (e.g., IFN-β, IFN-α), initiating their transcription. The primary outcome of the TRIF-dependent pathway is the production of type I interferons (IFN-α and IFN-β), which are essential for establishing an antiviral state.Retinoic acid-inducible gene-I (RIG-I)-like receptor (RLR) pathway [17,18,30,31]: Activated by RIG-I, MDA5, and LGP2, which are cytosolic sensors detecting viral RNA. Upon recognition of viral RNA, RIG-I and MDA5 interact with the mitochondrial antiviral signaling protein (MAVS, also known as IPS-1, VISA, or Cardif) located on the outer mitochondrial membrane. This interaction activates downstream signaling through TBK1 (TANK-binding kinase 1) and IKKε (IκB kinase epsilon), which leads to the activation of IRF3, IRF7, and NF-κB. The outcome is the production of type I interferons (IFN-α/β) and other pro-inflammatory cytokines, establishing an antiviral state.Cyclic GMP-AMP synthase (cGAS)–Stimulator of Interferon Genes (STING) pathway [32,33]: Activated by the cGAS (cyclic GMP-AMP synthase) receptor that is a cytosolic sensor detecting cytosolic DNA, which can originate from viruses (such as herpesviruses) or bacteria. Upon binding to cytosolic DNA, cGAS produces cGAMP, a second messenger that binds to the STING (Stimulator of Interferon Genes) protein located on the endoplasmic reticulum. STING then activates TBK1, leading to the phosphorylation and activation of IRF3/IRF7 and subsequent induction of type I interferons. This pathway is crucial for the detection of DNA viruses and triggers the production of type I interferons and other immune responses.The nucleotide-binding oligomerization domain (NOD)-like receptor (NLR) pathway [34,35]: The NOD-like receptor (NLR) pathway is a key component of the innate immune system, responsible for detecting intracellular pathogens and cellular stress. NLRs, including NOD1, NOD2, NLRP1, NLRP3, and NAIP, are cytoplasmic PRRs that recognize PAMPs and DAMPs, leading to the activation of inflammatory and immune responses. Although NOD1 and NOD2 mainly detect bacterial peptidoglycan fragments (e.g., muramyl dipeptide from bacterial cell walls), many RNA viruses, including influenza virus, dengue virus, and SARS-CoV-2, activate the NLRP3 inflammasome, leading to IL-1β production and exacerbated inflammation. Moreover, NOD2 also senses viral single-stranded RNA (ssRNA), leading to type I interferon production via activation of IRF3.

In summary, multiple signaling pathways, including the MyD88, TRIF, RIG-I/MAVS, cGAS-STING, and NLRP3 pathways, are involved in RNA virus-mediated PAMP-PRR signaling. These pathways coordinate the innate immune response, producing interferons and pro-inflammatory cytokines and establishing an antiviral or antimicrobial state in the host. We summarize the above-described pathways responding to viral infection in Table 1 and Figure 1.

This table outlines the key pathways activated by viral infection-caused PAMPs that lead to the transcription activation of inflammatory factors.

## 3. Virus-Induced Type I IFNs Stimulate Defensive Responses in Infected and Uninfected Cells by Activating ISGs

### 3.1. Introduction of Autocrine and Paracrine Signalings

Type I interferons (IFN-I), namely, IFN-α and IFN-β, generated from virus-infected cells, can defend cells against viral replication in both infected cells and uninfected cells. This defense mechanism occurs through both autocrine and paracrine signaling [42,43].

(1) In autocrine signaling, type I IFNs produced by an infected cell bind to the IFN-α/β receptors (IFNARs) on the surface of the same cell [44,45,46,47]. This binding triggers the JAK-STAT pathway [48,49,50,51,52], leading to the activation of interferon-stimulated genes (ISGs) within that cell. Some ISGs encode proteins that directly inhibit viral replication and promote antiviral defenses. For example, (1) viral double-stranded RNA (dsRNA) activates PKR (Protein Kinase R) and PKR inhibits protein synthesis by phosphorylating the translation initiation factor eIF2α, reducing viral replication [53]; (2) viral dsRNA activates OAS (2′-5′-Oligoadenylate Synthetase) that activates RNase L, which degrades viral and cellular RNA to halt viral replication [54,55]; and (3) viral RNA–protein complex activates Mx proteins [54]. These proteins inhibit the replication of several RNA viruses by trapping viral components and preventing their assembly [56,57,58]. This self-defense mechanism allows the infected cell to attempt to contain the viral infection and limits the virus’s ability to use the host machinery for replication.

(2) In addition to defending the infected cell, type I IFNs can also act in a paracrine manner. The secreted IFNs bind to IFNARs on neighboring, uninfected cells, activating the same JAK-STAT pathway and inducing the expression of ISGs in those cells [8,42,59]. This creates an antiviral state that prepares nearby cells to resist potential infection by (1) enhancing immune surveillance by promoting antigen presentation through MHC class I molecules, (2) blocking viral entry or uncoating in neighboring cells, and (3) amplifying other immune responses, such as recruiting immune cells like natural killer (NK) cells, which can directly kill infected cells. The paracrine defense is the only pathway used to prevent viral infection in uninfected cells. Thus, type I IFNs not only help the infected cell fight the virus but also establish a broader antiviral environment to limit the spread of infection within the tissue. Cells usually use the following nine pathways to promote some ISGs against viral infections.

### 3.2. Signaling Pathways Leading to Type I IFN-Mediated Antiviral Defense or Balance Antiviral Defense with Tissue Repair Processes

The establishment of the defending state of a cell via the IFNI-mediated pathway initiates with the interaction of IFNI and IFN-α/β receptors (IFNARs), which activates the JAK-STAT complex inside the cytoplasm. This, in turn, activates different downstream molecules that then enter the nucleus to activate the expression of ISGs. The following pathways are often presented in IFNI-IFNAR signalings (Figure 2).

(1) JAK-STAT Pathway [48,49,50,51,52]: The JAK-STAT pathway is the canonical signaling pathway activated by type I interferons (IFN-I). Upon IFN-I binding to its receptor complex (IFNAR1 and IFNAR2), conformational changes in the receptor lead to the recruitment and activation of the JAK kinases—JAK1 and TYK2. These kinases phosphorylate specific tyrosine residues on the cytoplasmic tails of IFNARs. This phosphorylation creates docking sites for the STAT proteins, particularly STAT1 and STAT2. Once recruited, these STAT proteins are phosphorylated by JAK1 and TYK2, leading to their dimerization. STAT1 and STAT2 form a heterodimer, which associates with IRF9 to create the ISGF3 (interferon-stimulated gene factor 3) complex. The ISGF3 complex then translocates into the nucleus, where it binds to ISREs (interferon-stimulated response elements) in the promoters of ISGs. This initiates the transcription of ISGs, leading to the antiviral state of the cell: ISGF3 binds to ISREs, a specific sequence in the promoter region of ISGs. This complex recruits co-activators and RNA polymerase II to initiate the transcription of ISGs, including antiviral effectors like Mx proteins, OAS1, and PKR, which limit viral replication by disrupting viral RNA synthesis or degrading viral RNA.

(2) Phosphatidylinositol 3-kinase (PI3K), serine/threonine protein kinase B (AKT), and mammalian target of rapamycin (mTOR) (PI3K-AKT-mTOR) Pathway [60,61,62]: The PI3K-AKT-mTOR pathway is critical for modulating cell survival, growth, and metabolism, and is activated in part by IFN-I. Activation of PI3K (phosphoinositide 3-kinase) by IFN-I is primarily via JAK-STAT signaling cascade and adaptor proteins. Briefly, IFN-I binds to the interferon-α/β receptor (IFNAR), a heterodimer composed of IFNAR1 and IFNAR2. This binding leads to the activation of JAK1 and TYK2, which phosphorylate the cytoplasmic domains of IFNAR1/2. Subsequently, IFNAR signaling recruits insulin receptor substrate ½ (IRS1/2) or \Grb2-associated binder proteins (Gab1/2). These adaptor proteins are phosphorylated by JAK kinases, creating docking sites for PI3K recruitment. Once recruited, class IA PI3K (p85-p110 heterodimer) binds to IRS1/2 or Gab1/2. The p85 regulatory subunit stabilizes PI3K, while the p110 catalytic subunit phosphorylates phosphatidylinositol 4,5-bisphosphate (PIP2) to generate phosphatidylinositol 3,4,5-trisphosphate (PIP3), a second messenger that recruits AKT (protein kinase B) to the plasma membrane. Once localized, AKT becomes activated through phosphorylation by upstream kinases such as PDK1 (3-phosphoinositide-dependent kinase 1). Activated AKT then phosphorylates downstream targets, including mTOR (mechanistic target of rapamycin). mTOR is a central regulator of protein synthesis through its ability to phosphorylate S6K and 4EBP1, which are crucial for ribosomal biogenesis and mRNA translation. mTOR enhances the translation of ISG mRNAs by modulating global protein synthesis machinery. The pathway fine-tunes the antiviral response by balancing cellular metabolism and the energy demands of mounting an immune response. By controlling the availability of translational machinery, mTOR ensures that critical ISGs, such as ISG15, IFITM family members, and IFIT proteins, are efficiently translated, further enhancing the antiviral state.

(3) MAPK Pathway (ERK, p38, JNK) [63,64,65,66]: Activated JAK-STAT signaling also triggers the activation of the mitogen-activated protein kinase (MAPK) pathway via recruiting adaptor proteins. The recruited adaptor proteins include growth factor receptor-bound protein 2 (Grb2) and Src homology, two domain-containing proteins (Shc) that are phosphorylated following IFNAR activation. These adaptors activate the small GTPases Ras or Rac1, which initiate distinct MAPK cascades: (1) Ras → RAF → MEK → ERK1/2; (2) Rac1 → MAP2K → JNK; and (3) ASK1 → MKK3/6 → p38 MAPK. The activation of ERK, JNK, and p38 MAPK leads to the phosphorylation and activation of downstream transcription factors, including AP-1 (Activator Protein-1), ATF2, and c-Jun. These transcription factors bind to promoter regions of ISGs and other immune response genes, promoting their expression. The MAPK pathway enhances ISG expression by activating transcription factors like AP-1, which binds to TRE (TPA-responsive element) and ISRE sites on the promoters of ISGs. This increases the transcription of genes involved in antiviral responses, such as MxA, IFIT1, and IFITM3, which interfere with viral replication, entry, and egress. p38 also contributes to the post-transcriptional regulation of cytokines and ISG mRNAs, enhancing their stability and translation.

(4) NF-κB Pathway [67,68,69]: IFN activates the NF-κB pathway via JAK-STAT signaling that activates the IKK complex (IκB kinase), which phosphorylates IκB proteins that normally sequester NF-κB in the cytoplasm. Upon phosphorylation, IκB is degraded by the proteasome, releasing NF-κB dimers (typically p50/p65) to translocate into the nucleus. NF-κB binds to specific sequences called κB sites in the promoters of target genes, including ISGs and pro-inflammatory cytokines like IL-6 and TNF-α. NF-κB enhances the transcription of ISGs by synergizing with other transcription factors such as IRF3/7 and AP-1. This cooperative action amplifies the antiviral response by inducing the production of some ISGs, including OAS, MxA, and PKR, which contribute to the degradation of viral RNA and inhibition of viral replication. NF-κB also regulates genes involved in inflammation and immune cell recruitment, thus bridging innate and adaptive immune responses.

(5) TGF-β Pathway: While typically associated with immune regulation and tissue repair, the transforming growth factor-β (TGF-β) pathway can interact with interferon signaling, particularly during chronic infections and inflammation. TGF-β ligands bind to the TGF-β receptors on the cell surface, leading to the phosphorylation of SMAD proteins (specifically SMAD2 and SMAD3), which then form a complex with SMAD4. This complex translocates into the nucleus, where it regulates the transcription of target genes. TGF-β signaling is often involved in modulating immune responses during chronic infections, inflammation, or tissue damage. Through SMAD-dependent transcriptional regulation, TGF-β can influence the expression of ISGs, particularly those involved in immune regulation, apoptosis, and fibrosis. Although TGF-β does not directly drive strong antiviral ISG activation like other pathways, it fine-tunes immune responses and can modulate long-term ISG activity to help balance antiviral defense with tissue repair processes. The interaction between TGF-beta and IFNs is two-fold: (1) IFN-I activates TGF-beta [70,71], and (2) TGF-beta inhibits IFN-gamma signaling [72]. Therefore, the role of TGF-beta in ISG-mediated antiviral responses needs to be investigated.

We summarize the above-described pathways that activate interferon-stimulated genes (ISGs) from type I interferons (IFN-I) in Table 2 and Figure 2.

This table outlines the key pathways activated by type I interferons (IFN-I) that lead to the transcription and expression of interferon-stimulated genes (ISGs), which are crucial for establishing an antiviral state within infected and uninfected cells. Each pathway interacts with various cellular components to ultimately enhance the immune response against viral and microbial pathogens.

In summary, type I interferons (IFN-α and IFN-β) protect infected and uninfected cells by establishing a robust antiviral state through ISG expression, shutting down viral replication mechanisms, enhancing immune surveillance, and preparing the host to rapidly counteract any viral invasion. This paracrine signaling is crucial for limiting the spread of infection and ensuring effective antiviral defense across tissues.

## 4. ISG15, ISGylation, and Their Role in Viral Infection and Replication

### 4.1. Introduction of Interferon-Stimulated Genes (ISGs)

ISGs are a diverse set of genes that are transcriptionally upregulated in response to type I (IFN-α/β), type II (IFN-γ), and type III (IFN-λ) interferons. More than 300 ISGs have been identified, which have been reviewed by Schoggins [73]. Some of these genes play crucial roles in the innate immune response by restricting viral replication, modulating immune signaling, and shaping adaptive immunity [74,75]. Based on their antiviral functions, ISGs can be classified into four categories (Table 3):

### 4.2. ISG15

Among the ISGs, we are particularly interested in ISG15 due to its role in ISGylation, a process functionally similar to ubiquitination. Ubiquitin (Ub) and Ubiquitin-like (Ubl) molecules are essential regulators of virus-mediated innate immunity. For example, Ub participates in the regulation of TNF-alpha-induced NFkB pathway activation, while ISG15, a Ubl, is involved in IFN-induced antiviral activity [107,108]. However, the precise mechanisms by which ISG15 modulates the antiviral functions of host cells remain less understood. The ISG15 gene consists of two exons and encodes a 17 kDa protein precursor that can be secreted from cells after production [109]. The newly translated ISG15 protein is composed of 165 amino acids (aa), and an 8 aa sequence at the C-terminus is cleaved to form the mature ISG15 protein. This mature ISG15 exposes a conserved 6 aa motif (LRLRGG) at its C-terminal end, which is also found in ubiquitin [110]. Viral infections trigger the production of type I interferons (IFNs), such as IFN-alpha and IFN-beta, which stimulate ISG15 production and promote ISGylation. In mice and cells deficient in type I IFN receptor R1 (IFNR1), ISG15 production decreases significantly following treatment with lipopolysaccharide (LPS) from Gram-negative bacteria or viral infections [111,112]. This suggests that stress-induced type I IFN secretion triggers ISG15 expression. However, a comprehensive understanding of ISG15’s role in viral infection remains elusive.

### 4.3. ISGylation

ISGylation is a process in which interferon-stimulated gene 15 (ISG15), a ubiquitin-like protein, is conjugated to target proteins at lysine residues. This process, induced by type I IFNs or indirectly by viral infections, involves a cascade of enzymatic reactions. This involves three sequential enzymatic steps:(1)The E1 enzyme, ubiquitin-like modifier activating enzyme 7 (UBA7, also known as UBE1L), binds to and activates ISG15 to expose its terminal LRLRGG sequences.(2)The E2 enzyme, ubiquitin-conjugating enzyme UBCH8 (also called UBE2L6), transfers ISG15 from UBA7 to the next step.(3)The E3 ligases, including HECT and RLD domain-containing E3 ubiquitin–protein ligase 5 (HERC5), ariadne RBR E3 ubiquitin–protein ligase 1 (ARIH1), and tripartite motif-containing protein 25 (TRIM25), facilitate the transfer of ISG15 from the E2 enzyme to the target protein, resulting in the conjugation of ISG15 to the ε-amino group of a lysine residue present in the target protein, completing the ISGylation process [113,114,115].

ISGylation is reversible, and this removal process, known as deISGylation, is catalyzed by ubiquitin-specific peptidase 18 (USP18), which cleaves ISG15 from ISGylated proteins [116,117,118]. The interactions between RNA viruses and the processes of ISGylation and deISGylation remain poorly understood, particularly as the roles of these two biological mechanisms have not been thoroughly investigated within viral infection systems.

### 4.4. ISGylation and Its Antiviral Mechanisms

The production of ISG15 is an essential part of the innate immune response, particularly in antiviral defense, and is tightly regulated by interferon signaling, including the production of ISG15. The interaction between ISG15 and various viral proteins, such as those from flaviviruses like Zika and West Nile, illustrates its complex role in promoting or inhibiting viral replication. ISGylation is also linked with innate immune pathways, including those mediated by RIG-I/MDA5 and TLR3 which coordinate the cellular response to viral threats [119,120,121]. ISGylation alters its substrate proteins’ function, stability, or localization, many of which are involved in key steps of viral replication and host immune responses [122]. It can enhance the degradation of viral proteins or interfere with viral assembly, limiting the efficiency of viral replication [119]. Furthermore, ISGylation of host proteins, such as those involved in interferon signaling, amplifies antiviral responses by modulating their activity. In addition to direct antiviral effects, ISG15 can also be secreted extracellularly, functioning as a cytokine to promote immune responses by stimulating the production of IFN-γ, a critical mediator of adaptive immunity [123].

### 4.5. Antiviral Role of ISG15 and ISGylation in Virus Infections and Viral Evasion of ISGylation and IFN Response

RNA viruses, such as influenza, hepatitis C virus (HCV), and coronaviruses, have evolved to interact with the ISG15 pathway. For example, the influenza B virus expresses a viral protein (NS1) that binds to ISG15, inhibiting its conjugation and thereby evading the host’s ISGylation-based antiviral defense [124]. Conversely, in certain infections, such as those caused by flaviviruses, ISGylation enhances the antiviral response by targeting viral proteins for degradation or disrupting viral replication complexes [125]. RNA viruses have developed diverse strategies to interact with and modulate the ISG15 pathway, either by evading or exploiting ISGylation to enhance their replication. Several key RNA viruses, including influenza, hepatitis C virus (HCV), coronaviruses, and flaviviruses, exhibit distinct mechanisms of interacting with the ISG15 system, highlighting the virus-specific nature of these interactions. Some viruses express deISGylating enzymes, which remove ISG15 from their viral proteins, thereby preventing ISGylation-mediated inhibition. Additionally, viruses like HIV and ZIKV can manipulate the host ISGylation machinery to facilitate their replication, highlighting the complex interplay between viruses and the host ISG15 pathway. Viral evasion mechanisms often involve direct interference with ISGylation, the removal of ISG15 from proteins, or the manipulation of host machinery to suppress ISG15’s antiviral effects. Some DNA viruses such as HBV and gammaherpesvirus are restricted by ISGylation.

#### 4.5.1. Influenza Virus

Influenza viruses, particularly influenza B virus (IBV), have evolved strategies to counteract ISGylation. IBV encodes the nonstructural protein 1 (NS1), which binds directly to ISG15. This interaction prevents ISG15 from being conjugated to its target proteins, thereby suppressing ISGylation-mediated antiviral responses. The inhibition of ISGylation by NS1 contributes to the virus’s ability to evade the host immune system and efficiently replicate [126]. Moreover, it has been shown that NS1 of influenza A virus (IAV) is less effective in binding ISG15, suggesting differences in how influenza virus subtypes manipulate the ISG15 pathway [127]. Studies in ISG15-deficient mice infected with IAV revealed increased viral load and more severe disease, emphasizing the protective role of ISGylation in controlling influenza infection [123,128]. In addition to ISGylation evasion, NS1 of both IAV and IBV suppresses type I IFN production. By targeting key signaling components like RIG-I and MAVS, NS1 reduces the host’s ability to produce interferons, effectively crippling the antiviral response. This suppression not only limits the upregulation of ISG15 but also prevents the activation of a broader range of ISGs, facilitating viral replication and dissemination within the host.

#### 4.5.2. Hepatitis C Virus (HCV)

HCV infection triggers a robust interferon response, leading to significant upregulation of ISG15 [129]. Furthermore, treatment with pegylated interferon (IFN) attenuates HCV replication [130]. However, HCV has developed sophisticated mechanisms to evade this immune defense. One critical viral protein, NS3/4A, is known to interact with and inhibit the antiviral activity of ISGylation [131,132]. NS3/4A serves as a protease that cleaves components of the host innate immune signaling pathway, including MAVS, a key adaptor protein involved in type I IFN production. In this way, HCV blocks ISG15-mediated antiviral responses, allowing the virus to persist and replicate in hepatocytes. Intriguingly, some studies have also demonstrated that HCV infection can induce ISGylation of viral proteins, affecting their function, stability, or interactions, though the exact consequences of these modifications remain a subject of ongoing investigation. HCV also expresses proteins that can interfere with ISGylation, though the precise mechanisms are still under investigation. By inhibiting ISGylation, HCV avoids being targeted by ISG15-mediated protein modification, which would otherwise impair viral replication. In this way, HCV exemplifies how viruses can employ both direct and indirect strategies to subvert ISGylation and IFN responses to sustain chronic infection.

#### 4.5.3. Coronaviruses

Coronaviruses, such as SARS-CoV and SARS-CoV-2, interact with the ISG15 pathway in complex ways. Recent studies have shown that ISG15 is significantly upregulated in response to SARS-CoV-2 infection, where it contributes to antiviral immunity by enhancing interferon production and ISGylating key proteins involved in viral replication [133]. However, coronaviruses also possess mechanisms to counteract ISGylation [133]. The papain-like protease (PLpro) of SARS-CoV and SARS-CoV-2 can act as a deISGylase, removing ISG15 from target proteins. This deISGylation activity dampens the host antiviral response, facilitating viral replication. The role of ISGylation in coronavirus infection is thus two-fold—while ISG15 is part of the innate antiviral defense, viral proteins such as PLpro can hijack this system to promote viral replication. This deISGylating activity also affects the host immune response by reducing the stability of key antiviral proteins, thus weakening the overall antiviral effect of ISGylation. In addition to its direct role in evading ISGylation, SARS-CoV-2 also modulates the interferon response to suppress ISG15 production. By inhibiting the induction of type I interferons and subsequent ISG expression, SARS-CoV-2 ensures that ISGylation does not effectively limit viral replication. The combination of deISGylation and interferon suppression enables SARS-CoV-2 to evade immune detection and propagate efficiently, contributing to the severity of COVID-19 in certain individuals.

#### 4.5.4. Flaviviruses (Zika and Dengue Virus)

Flaviviruses, including Zika virus (ZIKV) and Dengue virus (DENV), exploit the ISG15 system in different ways [90,134]. In these infections, ISG15 has been found to play a protective role by targeting viral proteins for ISGylation, leading to their degradation or functional impairment. For example, studies have demonstrated that ISGylation of the NS5 protein of ZIKV and DENV interferes with viral replication by destabilizing the viral replication complexes [135]. Furthermore, ISG15 has been shown to regulate other host factors involved in flavivirus replication, suggesting that its antiviral effects may be both direct (through ISGylation of viral proteins) and indirect (by modulating host immune factors). Importantly, some flaviviruses can manipulate the host ISGylation machinery to their advantage. For instance, DENV nonstructural proteins may antagonize ISGylation, although this mechanism is less well understood than in other RNA viruses. ZIKV has been shown to interfere with ISGylation to promote viral replication [134,136]. ZIKV expresses nonstructural proteins (NS1 and NS5) that interfere with host immune signaling pathways, including the ISG15 system [135,137]. Specifically, ZIKV NS5 has been found to inhibit interferon signaling by promoting the degradation of key immune signaling proteins like STAT2, thereby reducing ISG15 production and limiting its antiviral effects. In addition to inhibiting the IFN response, ZIKV directly targets the ISGylation pathway. Some studies suggest that ZIKV proteins can prevent ISG15 from conjugating with viral proteins, allowing the virus to escape ISGylation-induced inhibition. By reducing the overall efficacy of ISGylation and circumventing the immune response, ZIKV gains an advantage in replication and persistence, particularly in immune-privileged sites like the central nervous system and placenta.

#### 4.5.5. Ebola Virus

Ebola virus (EBOV) also highlights the versatility of ISG15 in antiviral defense. In EBOV infection, ISG15 is strongly induced, and its conjugation to host and viral proteins plays a key role in controlling viral replication [138,139]. ISGylation of viral proteins has been suggested to interfere with viral transcription and replication, although the specific viral targets of ISG15 in Ebola infection remain to be fully elucidated. EBOV’s VP35 protein, a known interferon antagonist, has been found to inhibit ISGylation indirectly by suppressing interferon production, thereby reducing ISG15 expression. This points to another layer of immune evasion employed by the virus to circumvent the antiviral effects of ISGylation. The viral protein VP35 is a potent interferon antagonist, capable of suppressing the activation of ISGs, including ISG15 [140,141]. By inhibiting the production of type I IFNs and preventing the full activation of the host antiviral response, VP35 allows EBOV to replicate unimpeded. The suppression of ISGylation helps EBOV evade immune surveillance, allowing the virus to persist in infected cells. This strategy reflects the virus’s ability to target multiple facets of the host immune response to evade ISG15-mediated antiviral defenses.

#### 4.5.6. Human Immunodeficiency Virus (HIV)

HIV-1 has several mechanisms to counteract the host’s antiviral defenses, including the manipulation of ISG15 and the interferon response [86,142]. Studies have shown that although HIV-1 infection upregulates ISG15, the virus circumvents ISG15’s antiviral effects by inhibiting ISGylation [143]. HIV’s viral protein R (Vpr) has been implicated in modulating host ubiquitination and ISGylation pathways, which suppresses the degradation of viral components. Additionally, HIV-1 proteins such as Nef and Tat play roles in modulating the IFN response, dampening the activation of ISGs (including ISG15), and limiting the host’s ability to mount an effective antiviral response. Moreover, the expression of ISG15 is thought to be involved in regulating type I IFN production in HIV infection. While ISG15 may enhance the immune response during the early stages of infection, HIV-1 manipulates host immune pathways to reduce sustained interferon signaling, allowing the virus to persist in latently infected cells. This manipulation highlights the dynamic interplay between viral evasion strategies and the host’s immune defense.

#### 4.5.7. Crimean–Congo Haemorrhagic Fever Virus (CCHFV)

CCHFV causes severe haemorrhagic fever in humans, with a mortality rate approaching 30%. Infection triggers a strong and rapid innate immune response, leading to the production of high levels of cytokines, including IFN and tumor necrosis factor-alpha (TNFα). To counteract ISGylation-mediated antiviral activity, CCHFV L protein decreases the levels of ubiquitinated and ISGylated proteins [108]. Notably, deISGylation of CCHFV L protein is dependent on its Ovarian Tumor (OTU) domain, which is conserved in many other viral proteins. Similar OTU domains are found in Arteriviruses, such as equine arteritis virus (EAV) and porcine respiratory and reproductive syndrome virus (PRRSV), where they reside within nonstructural protein 2 (nsp2) and also showed deISGylation activity.

#### 4.5.8. Gammaherpesviruses (MHV-68 and KSHV) and HBV

The antiviral activity of ISG15 and ISGylation is not limited to RNA viruses; ISG15-/- mice exhibited increased susceptibility to both herpes simplex virus type 1 (HSV-1) and murine gammaherpesvirus 68 (MHV-68), as well as Sindbis virus infection [107]. These findings indicate that ISG15 also functions as an antiviral molecule against DNA viruses. Notably, Kaposi’s sarcoma-associated herpesvirus (KSHV) vIRF1 has been reported to interact with the cellular ISG15 E3 ligase, HERC5, in the context of Toll-like receptor 3 (TLR3) activation and IFN induction, leading to a decrease in ISGylation [144]. Interestingly, hepatitis B virus (HBV) infection does not increase ISG15 expression. However, HBV replication is reduced by inhibition of UBP43 (USP18), as HBV DNA levels were significantly lower in the USP18-deficient mice compared to wild-type controls [145]. This suggests that ISGylation plays a role in HBV replication. There are conflicting reports regarding the levels of ISG15 upon HBV infection. Studies by Li et al. [146] and Kim et al. [145] reported that HBV infection does not affect ISG15 levels in hepatocellular carcinoma (HCC). Conversely, other studies have suggested a positive correlation between ISG15 expression and HBV infection [147]. Clinical data on ISG15 levels in HBV-infected patients have also been inconsistent [148,149]. Further research is needed to resolve these discrepancies and clarify the role of ISGylation in HBV infection.

Although ISG15 and ISGylation are believed to be crucial components of the host innate immune system, playing a dual role in directly inhibiting viral replication and modulating broader immune responses, it has also been reported that ISG15 promotes replications of HCV and ZIKV [134,150]. Their intricate relationship with viral infection and replication underscores the evolutionary arms race between host defenses and viral countermeasures. Understanding the precise mechanisms of ISGylation in various viral contexts is critical for the development of novel antiviral strategies.

## 5. Summary and Perspectives

This review has focused on the intricate interplay between viral PAMPs and host PRRs, which serves as a crucial trigger for initiating antiviral defense mechanisms. The recognition of viral PAMPs by PRRs activates intracellular signaling pathways that culminate in the production of type I IFNs and other pro-inflammatory cytokines. These signaling events lead to the induction of ISGs, a key group of antiviral effectors that establish an antiviral state within the infected cell and surrounding cells via autocrine and paracrine signaling pathways. Among the ISGs, ISG15 has emerged as a significant player in the host’s antiviral defense. ISG15 functions through a process called ISGylation, a ubiquitin-like modification of host and viral proteins that enhances the host’s ability to restrict viral replication. ISGylation affects various aspects of the viral life cycle, from inhibiting viral protein stability and function to modulating host immune responses that support the clearance of infected cells. Through these mechanisms, ISG15 and ISGylation represent a critical component of the innate immune response to viral infections.

However, the evolutionary arms race between viruses and host cells has driven the development of sophisticated viral evasion mechanisms. Viruses have evolved to counteract ISGylation and the broader interferon response, allowing them to evade host immunity and ensure their replication. This review has highlighted several viral strategies to evade ISG15-mediated responses, including the expression of deISGylating enzymes, direct inhibition of ISG15 conjugation, and interference with interferon signaling pathways. These mechanisms underscore the dynamic and continuous battle between host antiviral defenses and viral survival strategies.


**Future Directions:**


Looking ahead, several areas warrant further exploration:1.Understanding Virus-Specific Interactions with ISGylation:

While much progress has been made in elucidating the role of ISG15 and ISGylation in the context of specific viruses, a more comprehensive understanding of the molecular details governing these interactions is needed. The identification of viral proteins that ISGylation directly targets, and the mechanisms through which ISGylation disrupts viral processes, will be crucial for developing targeted antiviral therapies.

2.Exploring the Dual Role of ISG15 in Viral Infections:

Emerging evidence suggests that ISG15 may have dual roles during infection, potentially acting as both an antiviral factor and a proviral factor under certain conditions. Detailed investigations are required to clarify the conditions under which ISG15 exerts protective versus permissive effects on viral replication, and how viruses exploit these dual roles to enhance their survival.

3.Viral Evasion Mechanisms as Therapeutic Targets:

The ability of viruses to subvert ISGylation and other ISG-mediated responses represents a key challenge in antiviral therapy. A promising area of future research lies in targeting viral evasion proteins, such as deISGylating enzymes, to restore the efficacy of the host’s innate immune responses. Inhibitors of these viral proteins could be developed as novel therapeutic agents, capable of boosting the antiviral potential of ISGylation and other immune processes.

4.Enhancing ISGylation as an Antiviral Strategy:

Given the central role of ISG15 in antiviral defense, strategies to enhance ISGylation in infected cells could offer new avenues for therapeutic intervention. Understanding how ISGylation is regulated in different cell types and infection contexts may lead to the development of therapies that bolster the ISGylation machinery, potentially in combination with existing antiviral drugs or interferon therapies.

5.Therapeutic Potential of ISG15 as a Cytokine:

Beyond its role in ISGylation, ISG15 has been shown to act as a secreted cytokine, modulating the immune response through the induction of IFN-γ and other immune-stimulating factors. Exploring the therapeutic potential of ISG15 as a cytokine, its effects on various immune cells (i.e., NK cells, dendritic cells, etc.), and possible use in boosting immune responses in viral infections, cancer, and immunodeficiency disorders, remains a promising research avenue.

In conclusion, the PAMP-PRR interaction and subsequent ISG production, particularly ISG15-mediated ISGylation, represent a frontline defense in the host response to viral infections. Despite the viral evasion strategies that have evolved to counter these defenses, the continuing exploration of ISG15’s diverse roles in immunity holds great promise for the development of novel antiviral strategies. A deeper understanding of these interactions will provide important insights into both viral pathogenesis and the potential therapeutic manipulation of the innate immune system to combat viral diseases.

## Figures and Tables

**Figure 1 viruses-17-00528-f001:**
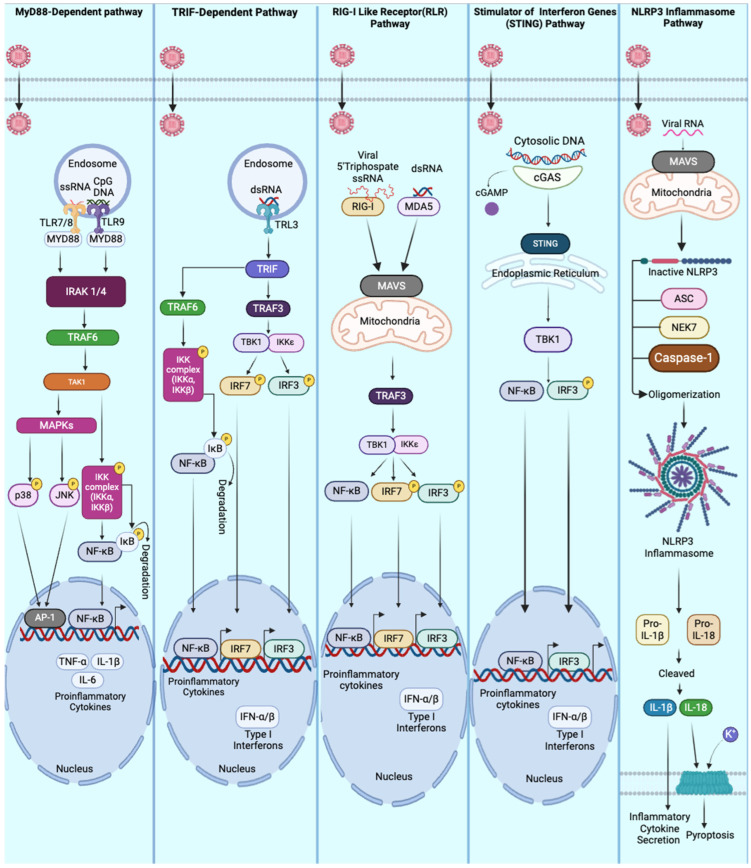
Innate immune pathways are activated during viral infections. This figure illustrates key cellular pathways involved in the innate immune response to viral infections. The MyD88-dependent pathway is initiated when viral PAMPs bind to Toll-like receptors (TLRs), leading to the recruitment of MyD88, IRAK1/4, and TRAF6. This activation triggers a series of MAPKs and NF-κB signaling cascades, resulting in the expression of pro-inflammatory cytokines. The TRIF-dependent pathway is activated by dsRNA binding to TLR3 in endosomes, initiating signals through TRIF to activate IRFs and NF-κB, leading to the production of type I interferons. The RIG-I-like receptor (RLR) pathway is triggered by the detection of viral RNA recognized by RIG-I or MDA5, which activates MAVS signaling at the mitochondria. This, in turn, stimulates IRFs and NF-κB to produce type I interferons and pro-inflammatory cytokines. The STING pathway is activated by cytosolic DNA recognized by cGAS. cGAS generates cGAMP molecules and activates STING in the endoplasmic reticulum, ultimately recruiting TBK1 to induce NF-κB and IRF activation. The NLRP3 inflammasome pathway is activated by viral RNA, leading to the assembly of the inflammasome complex, caspase-1 activation, and the secretion of pro-inflammatory cytokines.

**Figure 2 viruses-17-00528-f002:**
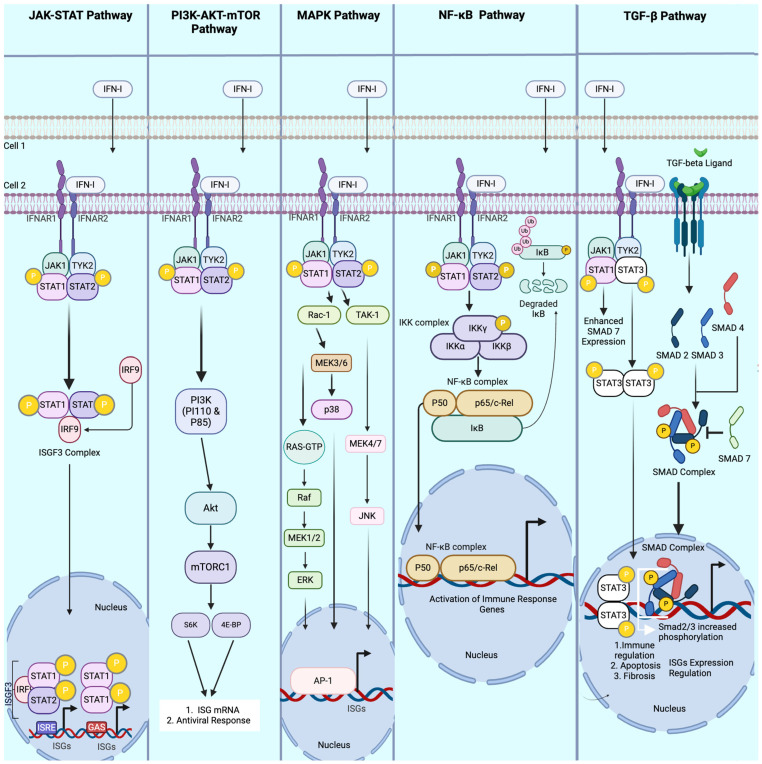
Key pathways modulated by Type I interferon (IFN-I) during viral infections. This figure illustrates the principal signaling pathways activated by IFN-I as it communicates antiviral signals from an infected cell to neighboring uninfected cells. The JAK-STAT pathways are initiated by IFN-I binding to IFNAR 1 and IFNAR 2 (interferon receptors), which, upon binding, recruits JAK 1 and TYK2. JAK 1 and TYK2 ultimately phosphorylate STAT 1 and STAT 2, respectively. STAT 1 and STAT 2 dimerize and form the ISGF3 complex with IRF9, which is translocated to the nucleus, inducing the expression of the interferon-stimulated genes (ISGs). IFN-I also modulates the PI3K-AKT-mTOR pathway, resulting in the ultimate expression of ISG mRNA and various antiviral responses. The MAPK pathways, activated by IFN-I, cause a series of signaling cases, such as p38, JNK, and Erk, all preceded by various signaling molecules (only three are depicted in this illustration). All three MAPK pathways induce ISG expression via AP-1 transcription regulation. Similarly, the NF-κB pathways induced by IFN-I are initiated by an IKK complex, recruiting the NF-κB complex activated by IκB to be translocated to the nucleus, driving the activation of immune response genes. Lastly, the TGF-β pathways are modulated by IFN-I through enhancing the signaling intensity of TGF-β receptors. A few examples of how this is achieved are (1) STAT 1 enhancement of SMA7 inhibitory protein expression, preventing the formation of the SMAD complex, and (2) STAT 3 dimerization and translocation to the nucleus increase SMAD2/3 phosphorylation, enhancing ISG expression regulation, immune regulation, apoptosis, and fibrosis.

**Table 1 viruses-17-00528-t001:** The key pathways activated by viral infection-caused PAMPs lead to the transcription activation of inflammatory factors.

Pathway	PRRs Involved	PAMPs Recognized	Key Adaptor Protein	Main Signaling Components	Main Outcome	References
**1. MyD88-Dependent pathway**	TLR7, TLR8,	ssRNA (TLR7/8)	MyD88	IRAK1/4, TRAF6, TAK1, NF-κB, MAPKs	Pro-inflammatory cytokine (TNF-α, IL-6, IL-1β) production	[24,25,26]
**2. TRIF-Dependent pathway**	TLR3	Viral dsRNA (TLR3)	TRIF	TRAF3, TBK1, IKKε, IRF3, IRF7, NF-κB	Type I interferon (IFN-α, IFN-β) production and some pro-inflammatory cytokines	[27,28,29]
**3. RLR Pathway**	RIG-I, MDA5	Viral 5′-triphosphate ssRNA (RIG-I), dsRNA (MDA5)	MAVS (mitochondrial antiviral signaling)	TRAF3, TBK1, IRF3, IRF7, NF-κB	Type I interferons (IFN-α/β) and pro-inflammatory cytokines	[31]
**4. cGAS-STING Pathway**	cGAS (Cyclic GMP-AMP synthase)	Cytosolic double-stranded DNA (dsDNA)	STING	TBK1, IRF3, NF-κB	Type I interferons (IFN-α/β), cGAMP production, antiviral responses	[36,37,38]
**5. NLRP3 Inflammasome**	NLRs, NOD2	Viral proteins (e.g., ORF3a, E and M of SARS-CoV-2, NS2A, M, E, and NS2B of DENV)	ASC (apoptosis-associated speck-like protein), RIP2	Caspase-1, IL-1β, IL-18, NF-κB, MAPK (JNK, p38)	Inflammasome activation, pyroptosis, IL-1β and IL-18 production	[35,39,40,41]

**Table 2 viruses-17-00528-t002:** The key pathways activated by type I interferons (IFN-I) that lead to the transcription and expression of interferon-stimulated genes (ISGs).

Pathway	Key Components	Mechanism of Action	ISG Activation Process
**1. JAK-STAT Pathway**	JAK1, TYK2, STAT1, STAT2, IRF9	IFN-I binds to IFNARs, activating JAK1 and TYK2, which phosphorylate STAT1/STAT2. These form the ISGF3 complex with IRF9, which translocates to the nucleus to activate ISG transcription.	ISGF3 binds to the IFN-stimulated response element (ISRE) in ISG promoters, driving their transcription.
**2. PI3K-AKT-mTOR Pathway**	PI3K, AKT, mTOR	IFN-I activates PI3K, leading to activation of AKT and mTOR, which modulates cellular metabolism and protein synthesis, indirectly influencing ISG expression.	mTOR activation affects translation of ISG mRNAs and enhances the antiviral response.
**3. MAPK Pathway (ERK, p38, JNK)**	ERK, p38, JNK	IFN-I activates MAPKs, which modulate transcription factors like AP-1 that are involved in ISG transcription.	MAPKs activate transcription factors that enhance the expression of a subset of ISGs involved in antiviral defense.
**4. NF-κB Pathway**	IKK complex, NF-κB	IFN-I induces the IKK complex, leading to the activation of NF-κB, which translocates to the nucleus and promotes ISG expression.	NF-κB binds to ISG promoters and synergizes with other transcription factors to activate ISG transcription.
**5. TGF-β Pathway**	TGF-β, SMADs	IFN-I can modulate TGF-β signaling, which, through SMAD proteins, influences ISG expression, particularly in immune regulation.	TGF-β/SMADs regulate ISG expression in response to tissue damage and chronic inflammation, enhancing long-term antiviral responses.

**Table 3 viruses-17-00528-t003:** ISG classification, function and targeted viruses.

Category	ISG Name	Function	Target Viruses	References
**Direct Antiviral Effectors**	**PKR (EIF2AK2)**	Phosphorylates eIF2α, inhibiting viral translation	Influenza, HCV, SARS-CoV-2	[76,77,78,79]
	**OAS1/RNase L**	Activates RNase L, degrades viral RNA	Influenza, HCV, SARS-CoV-2	[77,80,81]
	**MX1 (MxA) and MX2**	Blocks viral replication complexes	Influenza, HIV, VSV, HBV	[82,83,84]
	**ISG15**	Modifies host and viral proteins (**ISGylation**)	Influenza, SARS-CoV-2, HCV, RSV, DENV	[85,86,87,88,89,90]
	**ISG20**	Degrades viral RNA	Broad-spectrum	[91]
**Restriction Factors Against Specific Viruses**	**APOBEC3G**	Cytidine deaminase that mutates retroviral DNA	HIV	[92]
	**TRIM5α**	Binds retroviral capsids, blocking uncoating	HIV, Poxvirus	[93,94]
	**Tetherin (BST2)**	Prevents viral budding from the plasma membrane	HIV, Ebola, RSV, Influenza	[95,96,97,98]
**Immune Signaling Modulators**	**IRF1, IRF7, IRF9**	Regulate IFN production and ISG amplification	Broad-spectrum	Reviewed in [99]
	**USP18**	Negatively regulates IFN signaling	Broad-spectrum	[100]
	**SOCS Proteins**	Limit prolonged IFN responses	Broad-spectrum	[100,101]
**Host Metabolism and Cellular Regulation**	**Viperin (RSAD2)**	Disrupts lipid rafts and viral budding sites	Influenza, HCV	[102,103]
	**IFITM1, IFITM2, IFITM3**	Inhibits viral entry by altering membrane fluidity	Influenza, WNV, DENV, SARS-CoV-2	[104,105]
	**CH25H**	Produces 25-hydroxycholesterol (25HC), disrupting viral membrane fusion	Broad-spectrum	[106]

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
