# Peer review of "Chasing Virus Replication and Infection: PAMP-PRR Interaction Drives Type I Interferon Production, Which in Turn Activates ISG Expression and ISGylation"

_viruses, 2025, doi:10.3390/v17040528_

Round 1

Reviewer 1 Report (Previous Reviewer 1)

Comments and Suggestions for Authors

The manuscript have been revised accordingly to my previous critiques. This reviewer does not have further comments.

Reviewer 2 Report (Previous Reviewer 2)

Comments and Suggestions for Authors

The authors have addressed my previous concerns and have changed the organisation of the manuscript which is therefore now improved. 

Reviewer 3 Report (Previous Reviewer 3)

Comments and Suggestions for Authors

The authors did a nice job of addressing previous reviewer concerns.

This manuscript is a resubmission of an earlier submission. The following is a list of the peer review reports and author responses from that submission.

Round 1

Reviewer 1 Report

Comments and Suggestions for Authors

The manuscript have been revised accordingly to my previous critiques. This reviewer does not have further comments.

Reviewer 2 Report

Comments and Suggestions for Authors

The organisation of the review could be reorganized by the authors in order to have parts that flow logically and homogeneously.

For example, part III is organized as follows

I VIRUS-INDUCED TYPE I IFNS STIMULATE DEFENSIVE RESPONSES IN INFECTED AND UNINFECTED CELLS BY ACTIVATING ISGS

  1. Introduction of autocrine and paracrine signalings
  2. Signaling Pathways Leading to Type I IFNs-mediated antiviral defense.

                                    1) JAK-STAT Pathway

2) Phosphatidylinositol 3-kinase (PI3K), serine/threonine protein kinase B (AKT), and mammalian target of 212 rapamycin (mTOR) (PI3K-AKT-mTOR) Pathway

                                    3) MAPK Pathway (ERK, p38, JNK)

                                    4)NF-κB Pathway

                                    5) TGF-β Pathway

Part 1 could be an introduction and part 2 organized according to the different parts. The title of 2 "Signaling Pathways Leading to Type I IFNs-mediated antiviral defense." Could be revised to correspond better to the sub-sections (in particular TGF-beta, which does not correspond to this title).

Some of the references given by the authors are not related to the statements they are supposed to support. This should be reviewed to enable the information presented to be verified:

example :

-line 114: ref 53

Furthermore, in many paragraphs, the authors cite a large number of articles for a paragraph of just a few lines. It is difficult to see the point of citing so many articles without it being possible to really identify in the paragraph in question the information that was used in each article and that justifies it being cited. Authors should instead cite either the princeps article or a review article when describing a pathway, for example, which would provide readers with a reference to obtain more information if desired.

Part II

I suggest that authors list the PAMPs and PRRs in the first paragraph without trying to make links between who recognizes what, because trying to so in a few sentences usually leads to unfortunate shortcuts or to writing things that are simply wrong. For example, paragraphs 1-2 and 3 could be reduced to a few introductory sentences. In addition, the authors describe each pathway in the rest of the review, so readers interested in this level of detail need only refer to the corresponding paragraph. 

1) TLRs can recognize viral proteins (not only NLRs) - Zhou et al. 2021

The authors seem to be saying that NLRs are PRRs involved in the detection of viral proteins. To my knowledge, NLR do not recognize viral proteins, (and the reference that are provided do not support this), but rather DAMPs, even though some PRRs (but not proteins) can also be recognize.

Table 1: There is no legend, the sources should be cited here, the components described above are not all indicated (many TLRs are missing, for example). The PAMP recognized column is not correct because for NLRP3, these are not PAMPs.

Part III

Line 161 : Not all ISGs have antiviral properties, we should read "Some ISGs" instead of "ISGs".

On reading, it is not clear whether the authors wish to present the pathways activated by IFN or those that lead to the activation of ISGs; this should be specified (see TGF-beta remark below).

The authors describe several pathways, some of which are activated indirectly by type I IFN. They should ensure that the modalities of activation of the pathway in question are clearly described in each paragraph. This is missing at the very least for PI3K and MAPK (and could be clarified for NF-kb). I do not believe that TGF-beta is activated by IFN. Can the authors verify this information? The TGF-beta pathway is described more as an antagonist of the activation of certain ISGs.

MAPK pathway: it is mentioned that MxA, IFIT1, and IFITM3 are activated in response to MAPK activation. Does this apply specifically to these genes or more broadly to all ISGs?

Table 2: It is not relevant to indicate the PAMPS involved here (e.g. the jak-stat pathway is activated in response to IFN recognition, which is synthesized regardless of the PAMPs involved.

Part IV:

In the description of antiviral countermeasures against ISG-15, I don't think it's relevant to list all those that aim to inhibit ISG activation in one way or another, unless they specifically target ISG15. All viruses have countermeasures of this type, so it doesn't seem necessary to mention it, except in the introduction to this section, for example.

The table has no legend. The number of ISGs described to date should be mentioned somewhere, as the table only shows a fraction of them. 

Line 275: Are the 4 categories mentioned by the authors described classically or in other publications?

Line 286-292: this paragraph looks more like a review introduction than a sub-paragraph of a part IV (especially "This review aims to summarize the currently available data on virus-induced ISG15 production, the interaction of ISG15 with viral proteins leading to ISGylation, and the methodologies used in the study of viral ISGylation").

Lines 362-366: add a reference

Line 404: "though a filamentous virus", could the authors explain why this is important to note?

For CCHFV, it is "OUT" and not "OUT" for the field of ovarian tumours.

It would be interested in this Part (ISG15 section) to develop the molecular mechanisms that is involved, if different from the one presented in the introduction. More details about the viral countermeasure would be welcome.

Reviewer 3 Report

Comments and Suggestions for Authors

The authors did a very good job of addressing previous concerns and reviewer comments. The updated manuscript reads very well and will be an excellent resource to the scientific community.